# TextShield: Beyond Successfully Detecting Adversarial Sentences in text classification

**Lingfeng Shen**[1]* **Ze Zhang**[2] **Haiyun Jiang**[3]† **Ying Chen**[4]†
[1]Johns Hopkins University    [2]Tsinghua University    [3]Tencent AI Lab
[4]College of Information and Electrical Engineering, China Agricultural University
{starkshen3333,zezhang0826}@gmail.com
haiyunjiang1@tencent.com   chenying@cau.edu

## Abstract

Adversarial attack serves as a major challenge for neural network models in NLP, which precludes the model's deployment in safety-critical applications. A recent line of work, detection-based defense, aims to distinguish adversarial sentences from benign ones. However, the core limitation of previous detection methods is being incapable of giving correct predictions on adversarial sentences unlike defense methods from other paradigms. To solve this issue, this paper proposes TextShield: (1) we discover a link between text attack and saliency information, and then we propose a saliency-based detector, which can effectively detect whether an input sentence is adversarial or not. (2) We design a saliency-based corrector, which converts the detected adversary sentences to benign ones. By combining the saliency-based detector and corrector, TextShield extends the detection-only paradigm to a detection-correction paradigm, thus filling the gap in the existing detection-based defense. Comprehensive experiments show that (a) TextShield consistently achieves higher or comparable performance than state-of-the-art defense methods across various attacks on different benchmarks. (b) our saliency-based detector outperforms existing detectors for detecting adversary.

## 1 Introduction

Deep Neural Networks (DNNs) have obtained great progress in the field of natural language processing (NLP) but are vulnerable to adversarial attacks, leading to security and safety concerns, and research on defense algorithms against such attacks is urgently needed. Specifically, the most common attack for NLP is word-level attack (Wang et al., 2019b; Garg & Ramakrishnan, 2020; Zang et al., 2020; Li et al., 2021), which is usually implemented by adding, deleting or substituting words within a sentence. Such an attack often brings catastrophic performance degradation to DNN-based models. Therefore, we choose to study defending against word-level attacks[1] in this paper.

Although a number of defense methods can be found in the literature of NLP (Jia et al., 2019; Ko et al., 2019; Jones et al., 2020; Wang et al., 2020b; Zhou et al., 2021; Dong et al., 2021; Bao et al., 2021), there are remaining several unsolved research problems. One problem lies in the ineffective application of the existing detection-based defense paradigm to the adversarial defense scenario, which consists of two steps: adversarial detection that detects whether an input sentence is adversarial or not, and a model prediction that predicts a label for the input. Notice that the model prediction is usually specific for a practical application, e.g., different text classification tasks used in this paper. However, the detection-based defense paradigm only focuses on adversarial detection and does not make efforts to adversarial sentence prediction. That is, even if adversarial detection can successfully detect adversarial sentences, correctly predicting such adversaries remains incapable. Considering that an adversarial sentence is usually generated through adding an imperceptible perturbation into a benign sentence, we believe a practical defense paradigm should have another operation, a correction that transforms the adversarial sentence into the benign sentence as much as possible, and then

---

*This work is done during Lingfeng's internship at Tencent AI Lab, †refers to the corresponding authors.
[1]Through the rest of the paper, text attack specifically refers to word-level text attacks.

use the corrected adversarial sentence for the model prediction. Therefore, considering the paradigm problem, we propose a detection-correction defense method in this paper.

Another problem is that most existing defense methods rely either on word frequency or on prediction logits to detect adversarial sentences (Mozes et al., 2021; Mosca et al., 2022). One may wonder whether any other information can be leveraged for adversarial detection. In this paper, we discover that the mechanism of text attack is related to saliency, a measure used in the explainability of DNNs (Simonyan et al., 2013). Based on such discovery, we propose the saliency-based detector to enhance adversarial detection.

Overall, to address the two issues, in this paper, we build our TextShield, a defense method consisting of a detector and a corrector. Motivated by the link between adversarial sentence and saliency, we define a metric called adaptive word importance (AWI) based on saliency computation (Simonyan et al., 2013), and then design the detector and corrector based on AWI. Specifically, the detector combines four saliency-based sub-detectors to inform whether an input sentence is adversarial, where the four sub-detector utilize four saliency computation methods (Simonyan et al., 2013; Springenberg et al., 2015; Bach et al., 2015; Sundararajan et al., 2017) to calculate AWI, respectively. Then, if the detector recognizes the input as adversarial, the corrector rectifies the words with high adversarial probabilities in the sentence by their high-frequency synonyms, where the adversarial probability is determined by AWI. Finally, the corrected sentence is used for model prediction. Our extensive experiments demonstrate that TextShield can effectively defend against adversarial attacks on several text classification benchmarks.

Our main contribution are listed as follows:

- We design a defense method, TextShield, which extends the existing detection-based defense paradigm to detection-correction paradigm. The incorporation of a corrector in TextShield resolves the model prediction incapability problem in detection-based defense.

- We discover another word-level information for detecting adversarial sentences, saliency, and then design a detector based on adaptive word importance (AWI), a novel metric based on saliency computation. Furthermore, AWI is also used to design the corrector.

- Comprehensive experiments demonstrate that our TextShield performs comparably or better than SoTA defense methods under various victim classifiers on several text classification benchmarks.

## 2 RELATED WORK

In recent years, many adversarial attack methods have been proposed for NLP, different from backdoor attacks Chen et al. (2017); Liu et al. (2017); Wang et al. (2019a); Li et al. (2020c); Shen et al. (2022a), adversarial attacks try to modify the text data while making them less perceptible by humans. Coarsely, these attacks modifies the text data at character level (Belinkov & Bisk, 2018; Eger et al., 2019; He et al., 2021), word level (Alzantot et al., 2018; Zhang et al., 2021; Wang et al., 2022) or sentence level (Jia & Liang, 2017; Ribeiro et al., 2018; Zhang et al., 2019b; Lin et al., 2021). Since word-level text attack is the most common and effective attack for text classification, in this paper, we focus on defense methods against word-level text attacks, which can be categorized into three paradigms: (1) model-enhancement-based, (2) certified-robustness-based, and (3) detection-based.

Model enhancement is a common way to improve model's robustness against text attacks, and one popular way is adversarial training which re-trains a victim model with extra adversarial examples to obtain an enhanced model. Alzantot et al. (2018); Ren et al. (2019) generated adversarial examples by an attack method and re-trained models more robust against the attack. More successful defense based on model enhancement can be regarded as synonym encoding (Le et al., 2021; Dong et al., 2021; Wang et al., 2021a), a series of methods defend text attacks through setting the same embedding for different synonyms. Unfortunately, the robustness of re-trained models is often determined by the diversity of adversarial examples used in re-training (Shen et al., 2022b). As a result, adversarial training is vulnerable to unknown attacks.

Besides, a stream of recent popular defense methods (Huang et al., 2019; Jia et al., 2019) concentrates on certified robustness, which ensures models are robust to all potential word perturbations within the constraints. However, because of the extreme time cost in the training stage, certified ro-

bustness is more difficult to be applied to complex models compared to other paradigms of defenses. For example, IBP (Jia et al., 2019) performed catastrophically on large pre-trained language models.

Recently, some excellent works have designed detection-based defense methods. For example, Mozes et al. (2021) used the word frequency difference to design detectors; Le et al. (2021) designed a detector specific for the 'Universal trigger'; Mosca et al. (2022) proposed a detection method by leveraging prediction logits. Despite their success, such detection-based defense methods own a key issue: the incapability of correctly classifying adversarial sentences. An intuitive and straightforward idea is to design a corrector to convert adversarial sentences to their original benign ones, which has proved effective in the the field of computer vision (CV) (Liu et al., 2019; Qin et al., 2019; Li et al., 2020b; Deng et al., 2021). However, the different essence between languages and pixels makes correctors used in CV not transferable in NLP. In this paper, based on the generation mechanism of adversarial sentences, we design a proper detector and corrector for text attacks.

## 3 BACKGROUND

In this section, we present basic definitions used in adversarial text attack and saliency computation.

**Adversarial text attack** In a text classification scenario, $X = \{w_1, \ldots, w_d\}$ is a sentence with $d$ words, and $\mathbf{w}_1, \ldots, \mathbf{w}_d$ are the corresponding embeddings. Given a text classifier $F$, $F(X)$ denotes the probability distribution over the class set $\mathcal{Y}$ for input sentence $X$. Then, class $y_j$ with the largest probability in $F(X)$ is selected as the prediction of $X$. Furthermore, in a word-level text attack for a text classification scenario, there is a *victim classifier $F$* and a *text attack $G$*.

- Victim Classifier: Given a set of benign sentences in a text classification benchmark, the victim classifier $F$ can be trained with the benign sentences, where the sentence encoder is usually selected as TextCNN (Kim, 2014), LSTM (Hochreiter & Schmidhuber, 1997) or BERT (Devlin et al., 2019).

- Text Attack: For a benign sentence $X$ with true label $y_j$, the text attack $G$ generates an adversarial sentence $X^* = G(X)$ by substituting words $w_i$ in $X$ with $w_i^*$, satisfying: (1) $F(X^*)$ and $F(X)$ output different predicted labels; (2) $Dist(\mathbf{w}_i, \mathbf{w}_i^*) < \delta$, where $Dist(\cdot, \cdot)$ is a distance metric and $\delta$ is a threshold to limit the distance.

**Saliency computation** In the field of explainability of DNNs, *saliency map* is a common technique to identify input features that are most salient. In other words, these input features cause a maximum influence on the model's output. Specifically, *saliency* is a measure that quantifies the sensitivity of an output class for an input feature in a DNN-based model, and a saliency map is a transparent colored heatmap overlaid on the input features.

Furthermore, back-propagation-based methods, which have been widely used to compute saliency in the explainability of DNNs, treat saliency as the gradient of a DNN-based model. There are different ways to calculate gradients based on gradient signals passed from output to input during model training, which leads to different saliency computation methods. For example, in the vanilla gradient (VG) method (Simonyan et al., 2013), saliency is directly defined as the partial derivative of the network output concerning each input feature scaled by its value. Formally, given a classifier $F$ and a sentence $X$ containing words $w_i$, the saliency of $w_i$ to prediction $y_j$ is defined as follows:

$$r_{ij} = \frac{\partial F_{y_j}(X)}{\partial w_i} \tag{1}$$

where $F_{y_j}(X)$ is the prediction confidence for $y_j$ when applying $F$ on $X$.

## 4 MOTIVATION AND INTUITION

This section illustrates intuitions that motivate us to design TextShield. Firstly, we present our basic derivations to show a cross-cutting connection between adversarial robustness and saliency. Then, we define adaptive word importance (AWI), a metric that helps us to detect adversarial sentences.

### 4.1 THE CONNECTION BETWEEN ADVERSARIAL ROBUSTNESS AND SALIENCY

In this part, we explain why saliency is highly related to adversarial robustness, indicating there is a connection between adversarial robustness and saliency. Given a victim classifier $F$ and a benign sentence $X$ with label $y_j$ predicted by $F$, the objective of a text attack that generates an adversarial sentence $X^*$ with a wrong predicted label (namely $y_k$) is defined as follows:

$$\arg\min \mathcal{L}\left(F_{y_j}\left(X^*\right), y_k\right) \tag{2}$$

where $\mathcal{L}$ is a loss function (e.g., cross-entropy loss). In a vanilla gradient-based attack (Goodfellow et al., 2014), Eq 2 is optimized by a gradient descent method. Specifically, in each iteration of the attack, word $w_i$ is substituted with a word whose embedding is closest to $\mathbf{w_i}$. In other words, word embedding $\mathbf{w_i}$ is perturbed with a step rate $r_1$ by the attack, which is defined as follows:

$$w_i' = w_i - r_1 \frac{\partial \mathcal{L}(F_{y_j}(X), y_k)}{\partial w_i} \tag{3}$$

In ease of clarification, we regard the word embeddings as continuous variables. Then, based on Eq 1, we can further derive the term $w_i - r_1 \frac{\partial \mathcal{L}(F_{y_j}(X), y_k)}{\partial w_i}$ , and rewrite it as follows:

$$w_i - r_1 \frac{\partial \mathcal{L}(F_{y_j}(X), y_k)}{\partial F_{y_j}(X)} \cdot |r_{ij}| \tag{4}$$

As shown in Eq 4, words with larger $|r_{ij}|$ are perturbed more severely by the attack, indicating that saliency $|r_{ij}|$ may serve as a key for adversary detection.

### 4.2 ADAPTIVE WORD IMPORTANCE

Inspired from previous analysis and finding, we design a novel metric called adaptive word importance (AWI) based on saliency.

**Definition 1.** *Given sentence $X$ with label $y_j$ predicted by text classifier $F$, the adaptive word importance (AWI) of word $w_i$ in the sentence, $R_{ij}$, is defined as follows:*

$$R_{ij} = |\Delta(F, w_i, y_j)| \tag{5}$$

*$\Delta$ is a saliency computation method, e.g., VG.*

Given sentence $X$, the $(i, j)$-th element $R_{ij}$ reflects the importance of word $w_i$ towards prediction $y_j$ in victim classifier $F$. Thus, AWI can reflect the importance of a word, and a larger AWI value indicates higher importance.

## 5 TEXTSHIELD

### 5.1 OVERVIEW

Based on previous analysis, we present TextShield, *which is like a shield in front of the victim classifier*, as shown in Figure 1. Generally, TextShield has two principal components: a detector and a corrector. The detector is a learning-based binary classifier that takes a sentence as input and outputs whether the input is adversarial or not. The corrector continues to correct adversarial sentences to benign ones, which makes up the blank of previous detection-based defense methods.

Specifically, given an input sentence $X$, $X$ is fed to the victim classifier $F$ to get prediction $y_j$. Then the detector is used to judge whether $X$ is adversarial based on prediction $y_j$. If $X$ is benign, the final prediction for $X$ will be $y_j$ (i.e., $F(X)$). Otherwise, a corrector is used to transform $X$ to $X'$, and the final prediction for $X$ will be $F(X')$.

### 5.2 DETECTOR

As shown in Figure 2, the detector consists of four sub-detectors and one combiner. A sub-detector is a binary classifier that utilizes a saliency computation method and a LSTM layer to predict whether input sentence $X$ is adversarial or not. In this paper, four saliency computation methods, vanilla

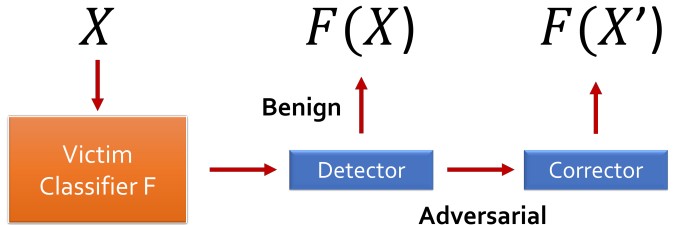

Figure 1: The overview of TextShield. During inference, a sentence $X$ is firstly fed to the classifier $F$. Then the detector judges whether $X$ is adversarial. If not, $F(X)$ will be the final prediction. On the contrary, $X$ will be transformed to $X'$ by a corrector, and the final prediction for $X$ is $F(X')$.

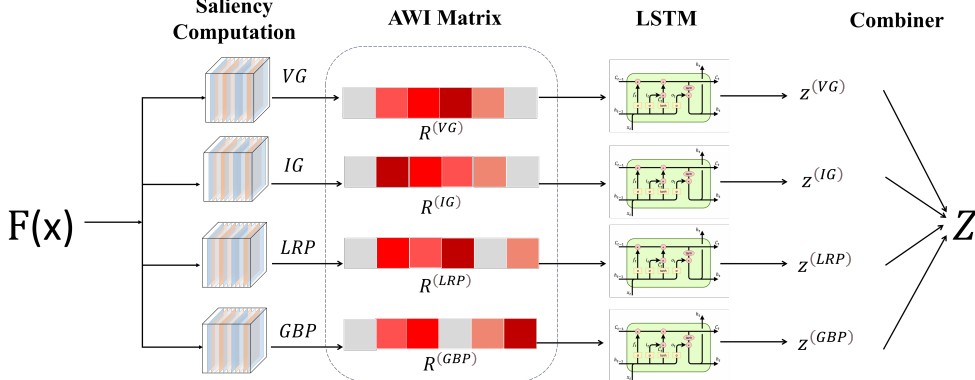

Figure 2: The overview of the detector. The detection can be summarized into three stages: (I): AWI matrices are generated by specific saliency computation methods based on $F(X)$ and the classifier $F$. (II): AWI matrices are fed to LSTMs to obtain predictions, e.g., $z^{(VG)}$, $z^{(GBP)}$, $z^{(LRP)}$ and $z^{(IG)}$. (III): The predictions are fed to a combiner (a linear layer) to produce a final prediction $z$.

gradient (VG), guided backpropagation (GBP), layerwise relevance propagation (LRP), and integration gradient (IG), are chosen, which correspond to the four sub-detectors, respectively. Then, the combiner uses a linear layer to combine four predictions from the four sub-detectors, respectively.

Specifically, for a sub-detector based on a saliency computation method $\mathcal{I}$, given an input sentence $X$, one AWI matrix is calculated for $X$ using $\mathcal{I}$, and then the AWI matrix is fed to an LSTM to obtain a prediction. Takings VG as an example, the VG-based sub-detector is calculated as follows.

$$z^{(VG)} = \text{LSTM}_1\left(\mathcal{R}^{(VG)}_{\cdot j}\right) \tag{6}$$

where $\mathcal{R}_{\cdot j}$ is a column of AWI matrix $\mathcal{R}$, and $z^{(VG)}$ is a binary prediction from VG-based sub-detector. Similarly, $z^{(GBP)}$, $z^{(LRP)}$ and $z^{(IG)}$ are predictions from GBP-based, LRP-based, and IG-based sub-detectors, respectively. Finally, through a linear combiner, the four binary predictions are combined to give the prediction $z$:

$$z = argmax(\text{L}([z^{(VG)}, z^{(GBP)}, z^{(LRP)}, z^{(IG)}]))$$

where L is the linear layer and $z$ is the label reflecting whether $X$ is adversarial.

In the following section, we will present the four saliency computation methods and then give the details of the detector's training.

**Vanilla Gradient (VG)**    For text classifier $F$, the AWI value of word $w_i$ in sentence $X$ to prediction $y_j$ is calculated by the vanilla gradient method (Simonyan et al., 2013) as follows:

$$\mathcal{R}^{(VG)}_{ij} = \frac{\partial F_{y_j}(X)}{\partial \mathbf{w_i}} \tag{7}$$

where $\mathcal{R}_{ij}^{(VG)}$, the $(i,j)$-th element of AWI matrix $\mathcal{R}^{(VG)}$, is the AWI value of word $w_i$ to prediction $y_j$, $F_{y_j}$ is the prediction confidence for $y_j$ from $F$, and $\mathbf{w_i}$ is the word embedding for word $w_i$. In fact, when calculating $\mathcal{R}_{ij}^{(VG)}$, $F_{y_j}$ is a scalar and $\mathbf{w_i} \in \mathbb{R}^{1 \times k}$ is a vector, where $k$ is the dimension of the word embedding. Thus, $\frac{\partial F_{y_j}(X)}{\partial \mathbf{w_i}}$ is a vector with the form $\mathbb{R}^{1 \times k}$, and an average operation is made to obtain $\mathcal{R}_{ij}^{(VG)}$. For brevity, we omit the operation in the paper.

**Guided Backpropagation (GBP)**    Following the idea of the vanilla gradient, GBP (Springenberg et al., 2015) chooses only 'positive' $(> 0)$ gradients and discards 'negative' $(< 0)$ gradients. Positive gradients indicate supporting the prediction, while negative ones mean the opposite. Such a choice enables us to capture what text classifier $F$ learns and yet discard anything that $F$ does not learn. We take the final outputs of GBP as AWI matrix $\mathcal{R}^{GBP}$.

**Layerwise Relevance Propagation (LRP)**    GBP owns an obvious drawback called gradient saturation (Bach et al., 2015; Binder et al., 2016) that gradients computed by GBP are always zeros under some cases. Therefore, Layerwise Relevance Propagation (LRP) (Bach et al., 2015) is proposed, which tackles the problem using a reference input besides the target input to calculate gradients. The LRP method calculates a weight for each connection in the gradient chain and uses a reference input instead of roughly setting the weights as 0 or 1 like GBP. The reference input is a 'neutral' input and will be different in different tasks (e.g., blank images in the CV field or zero embeddings in the NLP field). Finally, such gradients compose AWI matrix $\mathcal{R}^{LRP}$.

**Integrated Gradient (IG)**    LRP is calculating discrete gradients. However, the chain rule does not hold for discrete gradients so that LRP has essential drawbacks. Thus, the integrated gradient (IG) (Sundararajan et al., 2017) is proposed, which calculates the influence of word $w_i$ to prediction $y_j$ by integrating vanilla gradient, which is defined as follows:

$$\mathcal{R}_{ij}^{(IG)} = \left(\mathbf{w_i} - \mathbf{w_i}'\right) \times \int_0^1 \frac{\partial F_{y_j}\left(X, \mathbf{w_i}' + \alpha\left(\mathbf{w_i} - \mathbf{w_i}'\right)\right)}{\partial \mathbf{w_i}} \, \mathrm{d}\alpha \tag{8}$$

where $\mathbf{w_i}'$ is a zero embedding and $F_{y_j}(X, \mathbf{w_i}' + \alpha(\mathbf{w_i} - \mathbf{w_i}'))$ means that the embedding of word $w_i$, $\mathbf{w_i}$, is replaced with $\mathbf{w_i}' + \alpha(\mathbf{w_i} - \mathbf{w_i}')$. The final outputs of IG is AWI matrix $\mathcal{R}^{IG}$.

**Training of detector**    Generally, training the detector consists of the following three steps, where both the adversarial data setup and the balanced data setup prepare data for the model's training. More training details are deferred to Appendix A.

- **Adversarial data setup**: Given benign data, the victim classifier $F$ is fine-tuned by the data. Then the adversarial attacks are used to generate adversarial sentences and 2,000 adversarial sentences per class that successfully attacks are selected. Moreover, the adversarial sentences are equally generated by attacks.[2]

- **Balanced data setup**: The adversarial data and the same-size benign data are mixed as balanced data. Then, the balanced data is split into train-dev-test sets with a 7:2:1 proportion.

- **Model training**: The training data set is fed to train the detector, and the best-performance checkpoint is selected according to performance on the dev data set.

## 5.3    CORRECTOR

As mentioned in Sec 5.1, if the detector recognizes the input sentence $X$ as an adversarial one, then the prediction $y_j$ for $X$ is possibly inaccurate. Furthermore, as discussed in Sec 4.1, given an AWI matrix $\mathcal{R}$, larger $\mathcal{R}_{ij}$ means word $w_i$ is more important for classifying the sentence as $y_j$. In other words, words with large AWI values are more likely to be perturbed by attacks to fool the victim classifier. Therefore, the corrector corrects the words with large AWI values in the adversarial sentence. In addition, the corrector chooses $\mathcal{R}_{ij}^{VG}$ as $\mathcal{R}_{ij}$.

---

[2]**We do not train detector with the attack for evaluation. For example, if we evaluate the defense capacity against GA, then it is trained with IGA, PWWS, and TextFooler.**

Specifically, for an adversarial sentence $X^*$ with prediction $y_j$, the corrector regards a word $w_i$ as a suspect when its $\mathcal{R}_{ij}$ exceeds a threshold, which is defined as follows:

$$\mathcal{R}_{ij} > \beta \cdot (\mathcal{R}_{\max} - \mathcal{R}_{\min}) + \mathcal{R}_{\min} \tag{9}$$

where $\mathcal{R}_{\max}$ and $\mathcal{R}_{\min}$ are the largest and smallest values in $\mathcal{R}$, and $\beta$ is a hyper-parameter to control the ratio of suspects. For a suspect $w_i$, the corrector substitutes $w_i$ with the most frequent word $w' \in \mathcal{S}(w_i)$, where $\mathcal{S}(w_i)$ is $w_i$'s synonyms, extracted by NLTK (Loper & Bird, 2002).

## 6 EXPERIMENTS ON ADVERSARIAL DEFENSE

In this section, we conduct comprehensive experiments to compare TextShield to baselines.

### 6.1 EXPERIMENTAL SETTINGS

Firstly, we use **TextCNN** (Kim, 2014), **LSTM** (Hochreiter & Schmidhuber, 1997) and **BERT** (Devlin et al., 2019) as victim classifiers, and select four widely-used text attacks in the NLP field: **TL** (TextFooler, (Kuleshov et al., 2018)), **PWWS** (Probability Weighted Word Saliency, Ren et al., 2019) , **GA** (Genetic Algorithm, Alzantot et al., 2018) and **IGA** (Improved Genetic Algorithm, Wang et al., 2019c). Moreover, we choose three popular benchmarks in text classification: **IMDB** (Potts, 2010), **AG's News** (Zhang et al., 2015) and **Yahoo! Answers** (Zhang et al., 2015).

Then, given a victim classifier, an attack and a benchmark, we generate test data consisting of a benign dataset and an adversarial dataset, which are used to evaluate the generalization and robustness of defense methods. Specifically, the benign dataset includes 1,000 benign sentences sampled from the benchmark, and the adversarial dataset includes 1,000 adversarial sentences generated by the attack based on the benchmark. Please note that *there is no overlap between the test data and the training data used for the detector's training.* Finally, we evaluate the performance of the victim models with or without defense methods.

Moreover, we compare TextShield with four state-of-the-art defense methods: (1) **IBP** (Jia et al., 2019): the most representative method in the certified robustness paradigm. (2) **Adversarial Sparse Convex Combination (ASCC)** (Dong et al., 2021): a model-enhancement-based defense method which leverages convex hull to defend against adversarial word substitutions. (3) **Synonym Encoding Method (SEM)** (Wang et al., 2021a): a model-enhancement-based defense method based on synonym encoding. (4) **Frequency-based Word Substitution (FGWS)** (Mozes et al., 2021): a detection-based method which leverages word frequency. (5) **Word-level Differential Reaction (WDR)** (Mosca et al., 2022): a detection-based method which utilizes prediction logits. **Specifically, considering that FGWS and WDR are detection-only methods and cannot correctly classify adversarial sentences, we combine our corrector to them.** The baselines' settings are kept the same with corresponding papers.

### 6.2 PERFORMANCE

Table 1 lists the performances of various defense models under different scenarios, which demonstrates both the generalization and the robustness of these defense models.

**Effect on generalization** When evaluated on benign sentences (see Column 'No' in Table 1), normal training (NT) achieves the best results, and our proposed method, TextShield, achieves accuracy performance close to normal training (NT), indicating a good trade-off between robustness and generalization. By the way, such a trade-off is common for attack/defense, which has been theoretically investigated in the CV field (Zhang et al., 2019a). In addition, generalization can be seen as a proper upper bound for accuracy performance on adversarial sentences.

Moreover, among baselines, ASCC performs best, and IBP has lowest accuracy due to its relatively loose bounds. Besides, combined with our corrector, PGWS and WDR obtain comparable performances towards ASCC, which indicates that our corrector is effective across different detectors concerning model's generalization.

| Victim | Defense | IMDB | | | | | AGNews | | | | | Yahoo | | | | |
|---|---|---|---|---|---|---|---|---|---|---|---|---|---|---|---|---|
| | | No | IGA | PWWS | GA | TL | No | IGA | PWWS | GA | TL | No | IGA | PWWS | GA | TL |
| Text CNN | NT | 88.7 | 13.3 | 4.4 | 7.1 | 2.3 | 92.3 | 45.5 | 37.5 | 36.0 | 30.0 | 68.4 | 19.6 | 10.3 | 13.7 | 8.5 |
| | IBP | 78.6 | 72.5 | 72.5 | 71.5 | 65.0 | 89.4 | 80.0 | 80.5 | 80.5 | 80.0 | 64.2 | 49.4 | 52.6 | 59.2 | 61.0 |
| | FGWS | 84.7 | 70.4 | 68.0 | 70.2 | 64.0 | 84.7 | 82.0 | 83.5 | 82.2 | 83.4 | 65.0 | 57.0 | 58.0 | 57.5 | 60.8 |
| | WDR | 85.4 | 72.5 | 70.5 | 71.0 | 66.5 | 85.5 | 84.2 | 84.4 | 83.4 | 83.5 | 66.0 | 60.0 | 60.0 | 61.5 | 61.5 |
| | ASCC | 87.8 | 68.4 | 69.1 | 70.5 | 69.4 | 86.7 | 85.0 | 83.0 | 84.0 | 83.0 | 64.8 | 60.7 | 60.3 | 62.2 | 62.4 |
| | SEM | 86.8 | 66.4 | 71.1 | 72.5 | 72.4 | 89.7 | 86.0 | 86.0 | 85.0 | 86.0 | 65.8 | 61.0 | 60.5 | 63.0 | 63.0 |
| | TS(Ours) | 87.0 | 74.0 | 74.2 | 75.0 | 74.0 | 90.8 | 87.4 | 87.8 | 86.8 | 86.8 | 67.7 | 62.0 | 62.0 | 63.0 | 62.0 |
| LSTM | NT | 87.3 | 8.3 | 2.2 | 5.0 | 3.0 | 92.6 | 35.0 | 30.0 | 29.0 | 26.5 | 71.6 | 27.6 | 21.1 | 15.8 | 5.5 |
| | IBP | 79.5 | 70.0 | 70.0 | 69.0 | 64.5 | 86.3 | 79.5 | 79.5 | 76.5 | 79.5 | 60.2 | 35.0 | 33.5 | 30.0 | 31.5 |
| | FGWS | 80.9 | 70.4 | 71.1 | 72.9 | 70.0 | 87.5 | 82.5 | 82.0 | 82.0 | 82.0 | 68.0 | 58.0 | 58.2 | 58.5 | 58.5 |
| | WDR | 80.3 | 70.5 | 75.0 | 75.0 | 71.0 | 88.0 | 83.5 | 84.0 | 82.0 | 83.4 | 66.5 | 60.0 | 58.0 | 60.0 | 59.5 |
| | ASCC | 82.8 | 70.2 | 75.0 | 74.0 | 70.8 | 88.9 | 84.6 | 85.0 | 82.7 | 84.0 | 68.0 | 58.6 | 57.9 | 58.2 | 55.5 |
| | SEM | 86.8 | 72.2 | 77.0 | 77.0 | 73.8 | 90.9 | 85.5 | 86.5 | 85.0 | 85.5 | 69.0 | 60.6 | 59.9 | 60.2 | 62.5 |
| | TS(Ours) | 86.0 | 74.0 | 79.4 | 78.6 | 76.4 | 91.4 | 86.6 | 87.2 | 87.6 | 87.0 | 69.6 | 62.0 | 62.6 | 65.2 | 65.0 |
| BERT | NT | 92.3 | 24.5 | 40.7 | 40.0 | 28.0 | 94.6 | 66.5 | 68.0 | 58.5 | 45.5 | 77.7 | 31.3 | 34.3 | 15.7 | 10.0 |
| | IBP | - | - | - | - | - | - | - | - | - | - | - | - | - | - | - |
| | FGWS | 88.7 | 79.2 | 80.9 | 78.0 | 79.0 | 93.5 | 80.2 | 79.3 | 85.0 | 80.0 | 75.8 | 58.5 | 57.6 | 59.5 | 58.0 |
| | WDR | 88.5 | 82.0 | 82.0 | 82.5 | 81.5 | 93.0 | 82.5 | 84.0 | 86.5 | 83.5 | 76.5 | 61.8 | 60.8 | 62.4 | 60.6 |
| | ASCC | 88.2 | 84.3 | 84.2 | 84.2 | 82.0 | 90.1 | 83.5 | 83.5 | 87.7 | 87.6 | 74.5 | 60.7 | 62.5 | 63.5 | 61.1 |
| | SEM | 89.5 | 87.5 | 88.4 | 87.3 | 85.5 | 94.1 | 88.5 | 88.5 | 88.5 | 88.5 | 76.2 | 66.8 | 66.8 | 66.4 | 62.0 |
| | TS(Ours) | 88.3 | 88.4 | 88.6 | 90.5 | 86.5 | 94.0 | 90.2 | 90.0 | 90.0 | 89.5 | 76.0 | 70.5 | 69.0 | 70.0 | 64.5 |

Table 1: The accuracy (%) of various defense methods on benign and adversarial sentences. Rows (e.g., NT) represent the defense methods; Columns (e.g., No) represent the attacks that generate adversarial test sentences; No: no attack, NT: Normal Training, TS: TextShield, TL: TextFooler; BLUE is the best performance. Moreover, for a fair comparison, as the previous work (Shi et al., 2019) reported that BERT is too difficult to be tightly verified by IBP, we do not use IBP on BERT.

**Effect on robustness**   When evaluated on adversarial sentences, TextShield performs best in most cases, demonstrating that TextShield can effectively improve the model's robustness. Furthermore, the enhancement of model's robustness is derived from the leveraging a cross-cutting link between adversarial robustness and saliency, which is complementary to the conventional opinion (Zhang et al., 2020; Goyal et al., 2022) that text attacks should be better defended using discrete information instead of continuous information.

Moreover, among the baselines, SEM, the state-of-the-art defense method against word-level attacks, achieves the best performance. IBP, which was first proposed in the CV domain, is more suitable for TextCNN and does not perform excellently on other victim classifiers. FGWS focuses only on word frequencies, so it is not sufficient to detect adversarial sentences when word frequency is not distinct. Moreover, after combining with our corrector, WDR can yield a comparable performance towards ASCC, a competitive baseline using the model-enhancement-based paradigm, which indicates our corrector can combine well with existing detectors concerning model's robustness.

## 7   EXPERIMENTS ON ADVERSARIAL DETECTION

This section compares our saliency-based detector to current SoTA adversarial detectors in several configurations involving datasets and attacks.

### 7.1   EXPERIMENTAL SETTINGS

In the experiments, only **BERT** (Devlin et al., 2019) is used as victim classifier. Then, four widely used adversarial attacks are selected: **GA** (Alzantot et al., 2018), **PWWS** (Ren et al., 2019), **IGA** (Wang et al., 2019c) and **TextFooler** (Jin et al., 2020). Next, three popular benchmarks in text classification are selected: **IMDB** (Potts, 2010), **AG's News** (Zhang et al., 2015) and **Yelp** (Zhang et al., 2015). Similar to the experiments on adversarial defense, given a victim classifier, an attack, and a benchmark, we generate test data which include 1,000 adversarial sentences generated by the attack based on the benchmark. Finally, we evaluate the performance of the victim models on the test data. Following previous works (Mosca et al., 2022), the performance of adversarial detection is evaluated by F1-score and recall.

| Dataset | Attack | Saliency | | WDR | | FGWS | |
|---|---|---|---|---|---|---|---|
| | | F1-score | Recall | F1-score | Recall | F1-score | Recall |
| AGNews | PWWS | 92.2 ± 1.1 | 93.4 ± 0.4 | 90.1 ± 1.1 | 89.1 ± 1.4 | 88.4 | 80.1 |
| | IGA | 89.9 ± 1.2 | 90.4 ± 0.9 | 85.7 ± 1.3 | 86.6 ± 1.8 | 68.6 | 58.3 |
| | GA | 92.9 ± 1.0 | 91.4 ± 1.3 | 88.4 ± 0.9 | 87.2 ± 1.1 | 70.3 | 60.1 |
| | TextFooler | 95.1 ± 0.4 | 96.6 ± 0.7 | 94.0 ± 0.8 | 97.0 ± 0.9 | 86.0 | 77.6 |
| IMDB | PWWS | 92.7 ± 0.8 | 90.8 ± 1.0 | 89.2 ± 0.7 | 87.5 ± 1.8 | 80.8 | 76.7 |
| | IGA | 92.0 ± 1.2 | 93.2 ± 1.1 | 87.5 ± 0.9 | 90.5 ± 1.3 | 79.5 | 70.1 |
| | GA | 91.8 ± 0.9 | 92.1 ± 1.5 | 88.0 ± 1.1 | 89.6 ± 1.1 | 79.0 | 74.2 |
| | TextFooler | 96.6 ± 0.4 | 98.0 ± 0.6 | 95.0 ± 0.8 | 96.0 ± 0.9 | 86.0 | 77.6 |
| YELP | PWWS | 91.8 ± 1.5 | 90.1 ± 1.6 | 89.2 ± 1.4 | 86.2 ± 3.1 | 91.2 | 85.6 |
| | IGA | 91.0 ± 0.8 | 90.7 ± 1.0 | 87.1 ± 0.5 | 88.1 ± 1.9 | 85.4 | 82.8 |
| | GA | 91.2 ± 0.8 | 90.0 ± 0.8 | 87.2 ± 0.4 | 88.7 ± 1.4 | 85.9 | 83.3 |
| | TextFooler | 97.5 ± 0.3 | 98.3 ± 0.5 | 95.9 ± 0.3 | 97.5 ± 0.9 | 90.5 | 84.2 |

Table 2: The performances of adversarial detection under various attacks on BERT-based victim classifier. The performances of WDR and FGWS are from the corresponding paper; BLUE is the best performance. 'Saliency' represents our saliency-based detector.

Moreover, in order to examine the performance of our detector, two state-of-the-art detectors are chosen as baselines, **FGWS** (Mozes et al., 2021) and **WDR** (Mosca et al., 2022), where FGWS leverages word frequency to detect adversarial sentences, and WDR uses prediction logit information for adversarial detection.

## 7.2 PERFORMANCE

Table 2 shows the performance of the detectors in various configurations. In Table 2, our proposed detector consistently shows better results in terms of F1-score and recall. Specifically, our detector outperforms the best baseline in 11 configurations out of 12, and on average achieves 3.74% gain on F1-score. Such results show that the saliency-based detector reacts well to all selected attacks.

Moreover, we can also observe that TextFooler is easier to be detected than other attacks. Considering that TextFooler is much stronger than the attacks like GA and IGA, we can claim that there is no positive relation between attack's attack power and detection difficulty.

## 8 FURTHER RESULTS

We also illustrate the details or discussion towards critical designs of TextShield with additional experiments. We defer them to Appendix due to limited space. The outline is as follows: the analyses of hyper-parameters are shown in Appendix B. Performance against extra word-level attacks is shown in Appendix C. Also, human evaluation towards attack quality is shown in Appendix D. Besides, computational cost and corrector design discussions are shown in Appendix E and F. Adversarial detection performance of TextShield against other levels of attacks (e.g., sentence-level attack) is shown in Appendix G.

## 9 CONCLUSION

This paper proposes TextShield, a detect-correct pipelined defense method against word-level text attacks. Based on basic derivations, we find a cross-cutting connection between adversarial robustness and saliency, and then we propose the adaptive word importance (AWI) metric based on saliency computation. Based on AWI, we design a saliency-based detector, which achieves better performance than previous detectors. Then, we design a saliency-based corrector, which corrects adversarial sentences to benign ones using AWI. Comprehensive experiments demonstrate that TextShield is superior to previous defense methods for generalization and robustness.

## 10 ACKNOWLEDGEMENT

The research was financially supported by National Precision Agriculture Application Project (Grant/Contract number: JZNYYY001), and Beijing Municipal Science and Technology Project (Grant/Contract number: Z201100008020008).

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

## A  DETAILS OF DETECTOR'S TRAINING

The learnable parameters in our saliency-based detectors are the ones in the four LSTMs of the detector and the two-layer MLP of the combiner. After tokenization, we either conduct padding with 'max_length=128' or do a truncation for each input sentence. Then after saliency computation, we obtain the AWI matrix with a form $(128, |\mathcal{Y}|)$, where $|\mathcal{Y}|$ is the size of the label set used in the focused text classification. Thus, the four LSTMs take the four AWI matrices as input correspondingly and output their hidden states (hidden_size = 128). Finally, the hidden states are concatenated to feed to the combiner, which is a two-layer MLP + softmax function, with input_dim = 128, intermediate layer_dim = 64 and out_dim = 2. In addition, the LSTMs and the two-layer MLP are simultaneously trained through the Adam optimizer with a $5 \cdot e - 4$ learning rate.

## B  ABLATION STUDY

This section conducts ablation studies of TextShield to answer the following three questions: Q1) What is the impact of $\beta$, the hyper-parameters of the corrector? Q2) What is the impact of multiple sub-detectors? Q3) What is the impact of the number of adversarial sentences used to train detectors? The experimental settings are similar to the ones used in Sec 6.

## B.1 ANALYSIS OF HYPER-PARAMETER $\beta$

We study the impact of $\beta$, the hyper-parameters of our corrector. As shown in Figure 3, the best performance achieves when $\beta$ is 0.4, and either a small or large $\beta$ will cause a significant performance drop, especially when $\beta$ is large. This is because $\beta$ serves to control the ratio of suspects in a sentence: when $\beta$ is 0, only the most important word is considered as a suspect, which may be inadequate; on the contrary, when $\beta$ is 1.0, every word is viewed as a suspect, which is inappropriate.

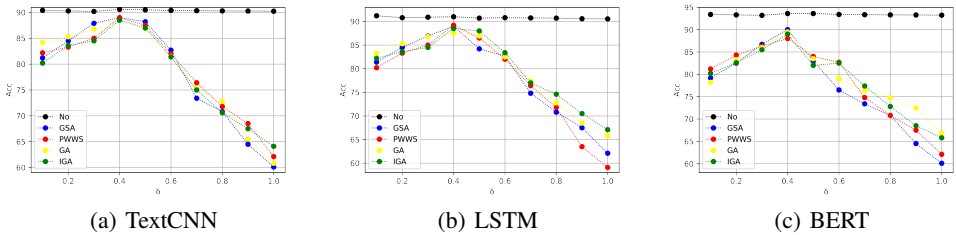

      (a) TextCNN            (b) LSTM            (c) BERT

Figure 3: The accuracy (%) of TextShield towards different text attacks when $\beta$ varies from 0.1 to 1.0. Specifically, the performance of the three victim classifiers on AGNews is presented.

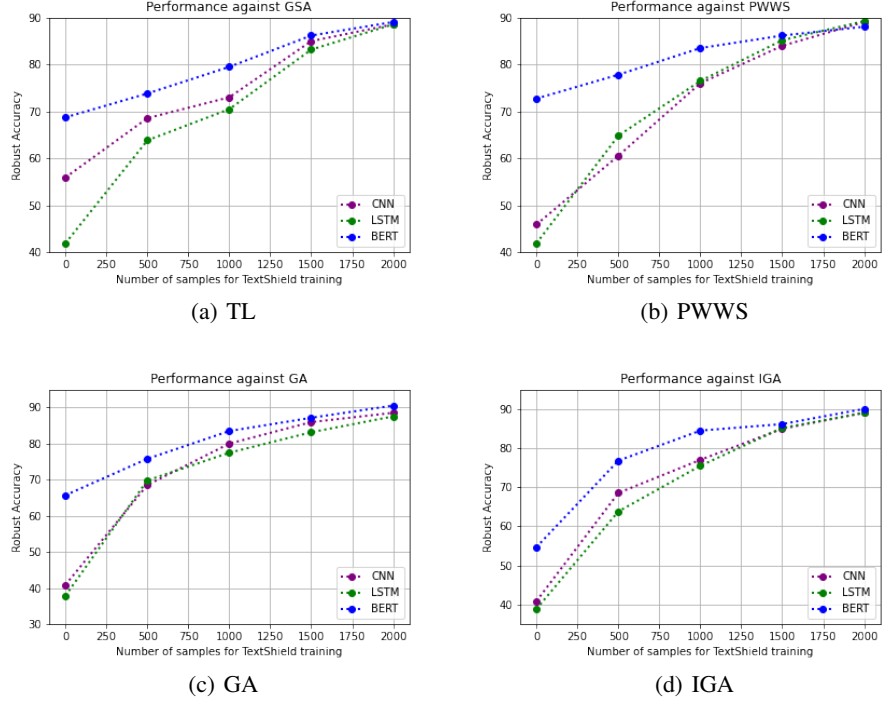

      (a) TL                    (b) PWWS

      (c) GA                    (d) IGA

Figure 4: Accuracy (%) of TextShield on adversarial sentences generated by four attacks (TL, PWWS, GA and IGA) as $K$ changes.

## B.2 ANALYSIS OF THE COMBINING OF MULTIPLE SUB-DETECTORS

Since a crucial design of TextShield is the ensembling of the four sub-detectors with a combiner, we investigate the impact of such an ensembling strategy. Specifically, we verify the performance after removing some sub-detectors, and the evaluation is launched using TextCNN on AGNews.

As shown in Table 3, removing any sub-detector will cause a performance drop, demonstrating the effectiveness of each sub-detector. Based on the results, the importance of sub-detectors can

be ranked as follows: VG>LRP>IG>GBP. Furthermore, removing all the sub-detectors shows a catastrophic performance drop, which confirms our arguments that different sub-detectors can complement well with each other and also shows the importance of integrating the four sub-detectors.

Please note that neither the saliency computation methods nor the ensembling strategy of various sub-detectors are our paper's focus. There are more sophisticated methods for saliency computation and sub-detector combination, and such methods can also be applied to TextShield.

| Removed sub-detector | TL | PWWS | GA | IGA |
|---|---|---|---|---|
| -VG sub-detector | -8.9 | -7.9 | -6.4 | -7.0 |
| -IG sub-detector | -6.8 | -4.5 | -3.0 | -4.5 |
| -LRP sub-detector | -6.6 | -5.0 | -2.8 | -4.0 |
| -GBP detector | -7.4 | -2.5 | -4.1 | -5.1 |
| -All sub-detectors | -34.8 | -29.6 | -40.3 | -36.9 |

Table 3: Accuracy (%) drop on adversarial sentences generated by different attacks after removing some sub-detectors from TextShield. E.g., '-VG' means removing the VG-based sub-detector in TextShield.

### B.3 ANALYSIS OF THE NUMBER OF ADVERSARIAL SENTENCES USED IN TRAINING DETECTORS

During the training process, we use the same number $K$ of adversarial sentences per class, where $K = 2000$. This experiment illustrates the relation between the performance of TextShield and $K$. Notice that, except for $K$, we keep the same training settings as the experimental setting used in Sec 6. The performance of TextShield under different $K$ is shown in Figure 4.

As illustrated in Figure 4, the defense capacity of TextShield is enhanced as $K$ continuously increases since more samples used in the training data help the detector better recognize adversarial sentences. Specifically, when $K$ is larger than 2000, the improvement becomes extremely subtle, indicating that we have arrived at the upper-bound of TextShield performance under such configurations.

| Dataset | BERT | | | | | | |
|---|---|---|---|---|---|---|---|
| | NT | IBP | FGWS | WDR | ASCC | SEM | TS |
| IMDb | 10.8 | - | 50.2 | 59.0 | 61.4 | 61.4 | 64.5 |
| AGNews | 9.6 | - | 46.7 | 52.4 | 60.2 | 59.6 | 63.0 |
| Yahoo | 6.0 | - | 40.2 | 50.1 | 58.9 | 60.3 | 64.2 |

Table 4: The accuracy (%) of various defense methods towards BERT-Attack (BA).

## C PERFORMANCE ON OTHER WORD-LEVEL ATTACKS

Additionally, we add BERT-attack (BA) Li et al. (2020a) to our attack methods. The settings of BA are the same from its paper Li et al. (2020a). Specifically, please note that we also train our detectors with adversarial samples generated by BA (same sample size as the experiments in Sec 6). Then, we measure the defense capacity for each baseline, and the results are shown in Table 4.

## D HUMAN EVALUATION FOR THE TEXT ATTACK QUALITY

The quality of text attacks remains a challenge since it is difficult to ensure that text attacks will keep sentences' ground truth towards the original label. **This is a problem that many SoTA adversarial defense methods have omitted.** Therefore, we conduct a human evaluation by examining the quality of four attacks selected in Sec 6. Considering there are over one thousand adversarial samples, which lead to the tremendous manual effort, we thus choose to sample 200 adversarial sentences for

|  | GA | IGA | PWWS | TextFooler |
|---|---|---|---|---|
| Quality | 96.4 | 94.3 | 95.0 | 93.0 |

Table 5: The quality of five text attacks selected in our paper. Quality represents the proportion of adversarial sentences whose ground-truth remain unchanged. We can see, in most cases, text attacks do not affect the ground-truth.

our selected attacks. Then, we evaluate whether such attacks modify their ground-truth labels, and the results are illustrated in Table 5. Then, we also present the F1-score on these samples (conducted human evaluation by us) in Table 6. We can observe that the overall quality of adversarial sentences is good.

| Victim | Defense | IMDB | | | | | AGNews | | | | | Yahoo | | | | |
|---|---|---|---|---|---|---|---|---|---|---|---|---|---|---|---|---|
| | | No | IGA | PWWS | GA | TL | No | IGA | PWWS | GA | TL | No | IGA | PWWS | GA | TL |
| Text CNN | NT | 88.7 | 12.3 | 4.5 | 7.7 | 3.3 | 92.0 | 43.5 | 35.5 | 34.5 | 30.0 | 67.5 | 19.0 | 10.9 | 14.8 | 9.5 |
| | IBP | 76.6 | 71.2 | 71.2 | 70.0 | 62.5 | 89.0 | 79.2 | 80.2 | 80.3 | 79.6 | 62.4 | 49.0 | 51.4 | 58.2 | 60.2 |
| | FGWS | 83.8 | 70.0 | 67.2 | 68.8 | 63.0 | 83.5 | 81.8 | 83.0 | 81.2 | 82.2 | 64.1 | 56.1 | 57.2 | 56.5 | 58.7 |
| | WDR | 85.0 | 71.4 | 69.2 | 70.0 | 64.4 | 83.8 | 82.3 | 83.1 | 82.0 | 82.4 | 65.0 | 59.1 | 58.5 | 60.0 | 60.2 |
| | ASCC | 87.5 | 66.9 | 68.2 | 69.4 | 68.2 | 86.7 | 84.0 | 81.8 | 82.2 | 81.6 | 63.2 | 59.8 | 60.0 | 60.5 | 61.0 |
| | SEM | 87.0 | 66.8 | 72.0 | 71.0 | 71.6 | 89.6 | 85.2 | 85.0 | 84.4 | 85.2 | 63.9 | 60.0 | 60.5 | 61.0 | 61.7 |
| | TS | 87.0 | 73.2 | 73.6 | 74.2 | 73.5 | 90.2 | 87.0 | 87.5 | 85.9 | 86.2 | 67.0 | 61.4 | 58.3 | 62.2 | 61.4 |
| LSTM | NT | 87.0 | 8.9 | 4.8 | 7.0 | 4.0 | 92.4 | 34.0 | 30.0 | 29.8 | 25.4 | 71.0 | 27.2 | 21.0 | 15.1 | 6.5 |
| | IBP | 78.2 | 69.0 | 68.0 | 65.8 | 61.5 | 85.6 | 78.4 | 78.6 | 75.5 | 78.8 | 60.0 | 34.1 | 32.8 | 29.1 | 30.3 |
| | FGWS | 80.2 | 69.6 | 70.3 | 71.9 | 69.0 | 86.8 | 81.7 | 81.1 | 81.2 | 80.8 | 66.8 | 57.0 | 57.5 | 57.8 | 57.4 |
| | WDR | 80.1 | 70.1 | 74.1 | 73.8 | 69.7 | 86.4 | 82.6 | 83.5 | 81.1 | 82.6 | 65.2 | 59.0 | 57.5 | 58.9 | 58.6 |
| | ASCC | 81.7 | 69.5 | 74.0 | 73.1 | 70.1 | 86.9 | 83.7 | 84.0 | 81.4 | 82.8 | 67.0 | 57.8 | 56.9 | 57.4 | 54.7 |
| | SEM | 85.9 | 71.4 | 76.0 | 75.8 | 72.9 | 90.1 | 84.3 | 85.8 | 84.1 | 84.3 | 68.2 | 60.0 | 58.7 | 59.4 | 61.2 |
| | TS | 85.8 | 73.2 | 78.8 | 78.0 | 75.8 | 90.6 | 85.8 | 87.2 | 87.1 | 86.3 | 69.2 | 62.0 | 61.9 | 64.7 | 63.9 |
| BERT | NT | 92.0 | 25.4 | 36.9 | 39.1 | 26.8 | 94.0 | 65.7 | 67.1 | 57.6 | 44.8 | 76.9 | 30.7 | 33.6 | 16.5 | 11.6 |
| | IBP | - | - | - | - | - | - | - | - | - | - | - | - | - | - | - |
| | FGWS | 88.8 | 79.7 | 80.3 | 78.7 | 79.3 | 93.2 | 79.6 | 79.0 | 84.3 | 80.5 | 76.0 | 56.9 | 58.0 | 58.4 | 58.3 |
| | WDR | 89.2 | 81.2 | 81.0 | 81.6 | 81.6 | 92.7 | 82.4 | 84.2 | 86.1 | 83.2 | 76.0 | 60.3 | 60.2 | 61.0 | 60.0 |
| | ASCC | 88.0 | 83.6 | 83.8 | 83.5 | 81.0 | 90.0 | 82.7 | 82.8 | 87.0 | 86.3 | 73.3 | 60.0 | 61.6 | 62.3 | 60.3 |
| | SEM | 89.9 | 86.4 | 87.5 | 86.0 | 85.0 | 94.1 | 88.0 | 87.9 | 88.5 | 89.2 | 75.8 | 66.2 | 65.9 | 66.0 | 62.2 |
| | TS | 88.4 | 87.9 | 88.0 | 90.5 | 85.8 | 92.0 | 89.8 | 89.6 | 89.3 | 89.0 | 75.2 | 67.6 | 67.2 | 69.1 | 63.7 |

Table 6: Under human-evaluated adversarial sentences, F1-score (%) of various defense methods on adversarial sentences.

# E  COMPUTATIONAL COST

Concerning the computational cost of our method, we present a case study here. On one RTX3090 GPU, the victim model is selected as BERT-base-uncased. Setting the batch size as 8, each inference of BERT takes around 20-30ms, TextShield (only detector part) takes around 60ms, and the cost is acceptable. The corrector + BERT inference takes around 20-30ms, nearly equal to BERT inference time since the information corrector needed has already been computed in the VG detector. Overall, it takes around 80-90ms per inference time under the configuration. Considering the training cost of detectors, since detectors only consist of LSTM and MLP, training them until convergence takes around 60min (On RTX3090).

# F  OTHER BASELINES FOR CORRECTOR DESIGN

Since the baselines for our corrector remain blank, we define some intuitive strategies for the corrector as follows: (1) replace words based on their part of speech (POS), (2) we replace words based on their word frequency in the benchmark (Freq). The only thing that varied in this set of experiments is: how we select words to be replaced by high-frequency synonyms. Specifically, since different selection methods have different optimal $\beta$, we tune $\beta$ for these two baselines and report the best performance. The remaining settings in the corrector are the same to ensure fair comparisons. The results are shown in Table 7. We can see that all of the methods perform significantly worse than

our original design - leverage saliency for correction. Specifically, we can see that combining verbs and nouns outperform merely either of them, which is reasonable. Also, we find that replacing low-frequency words achieves a not bad performance, which matches the empirical findings in Mozes et al. (2021). Overall, the results demonstrate the effectiveness of our corrector design.

| Detection | TL | PWWS | GA | IGA |
|---|---|---|---|---|
| POS(Verb) | -14.9 | -13.2 | -16.4 | -14.7 |
| POS(Noun) | -12.8 | -10.4 | -9.0 | -10.3 |
| POS(Noun+Verb) | -6.7 | -7.0 | -5.8 | -6.0 |
| Freq(Low) | -8.9 | -10.6 | -7.3 | -9.9 |

Table 7: Accuracy (%) drop on adversarial sentences by setting different replacement strategies in the corrector. POS(Verb), POS(Noun), POS(Noun+Verb) mean only replace the verbs, nouns, verb+noun with their high-frequency synonyms. Freq(Low) represent replace low-frequency word with their high-frequency synonyms.

## G  LIMITATION AND FUTURE WORKS

TextShield mainly focuses on defending against word-level text attacks, so the main limitation in TextShield lies in the specificity of word-level attacks. Moreover, this limitation also exists in most adversarial defenses. To validate the practical application of TextShield, in this section, we conduct similar experiments as the ones in Sec 7 yet on detecting sentence-level and character-level attacks (Wang et al., 2021a; Dong et al., 2021; Mozes et al., 2021; Mosca et al., 2022). Specifically, we select one character-level attack, Visual  (Eger et al., 2019), and two sentence-level attacks, PAWS (Zhang et al., 2019b) and T3 (Wang et al., 2020a), and show the results in Table 8.

| Dataset | Attack | Saliency | | WDR | | FGWS | |
|---|---|---|---|---|---|---|---|
| | | F1-score | recall | F1-score | recall | F1-score | recall |
| AGNews | PAWS | 80.9 ± 1.0 | 80.5 ± 0.6 | 73.4 ± 1.0 | 74.3 ± 1.1 | 68.7 | 67.0 |
| | T3 | 76.9 ± 1.2 | 76.8 ± 0.9 | 73.0 ± 1.3 | 71.5 ± 1.2 | 60.4 | 60.3 |
| | Visual | 84.9 ± 1.0 | 83.8 ± 0.8 | 80.0 ± 1.1 | 78.4 ± 0.8 | 70.1 | 68.5 |
| IMDB | PAWS | 80.3 ± 0.6 | 80.4 ± 1.1 | 75.3 ± 0.8 | 72.3 ± 1.0 | 65.4 | 63.8 |
| | T3 | 81.4 ± 1.1 | 80.3 ± 1.4 | 74.7 ± 0.8 | 74.4 ± 1.8 | 68.4 | 66.0 |
| | Visual | 84.9 ± 1.0 | 85.3 ± 0.4 | 80.6 ± 0.6 | 79.0 ± 1.1 | 74.5 | 71.1 |
| YELP | PAWS | 80.4 ± 1.0 | 80.1 ± 0.6 | 77.2 ± 1.2 | 74.2 ± 2.1 | 70.2 | 71.6 |
| | T3 | 80.1 ± 0.6 | 81.7 ± 0.6 | 75.0 ± 0.6 | 72.5 ± 1.6 | 70.0 | 69.7 |
| | Visual | 85.1 ± 0.4 | 85.0 ± 0.8 | 81.7 ± 0.8 | 80.2 ± 1.1 | 78.0 | 74.7 |

Table 8: The adversarial detection performance of different detectors against sentence-level and character-level attacks. Generally, all the detectors become less effective when facing sentence-level attacks, but our saliency-based detector still outperforms the baselines.

First, we can observe that all the detectors have a significant performance drop (about 10%) in both F1-score and recall when facing sentence-level attacks, compared to the performance on detecting word-level attacks in Table 1. The reason might be that the perturbation in sentence-level attacks is more complicated and is more likely to influence sentence' semantics at a level higher than word-level attacks, which makes sentence-level attacks more difficult to detect. Moreover, the detection difficulty of character-level attacks lies between word-level attacks and sentence-level attacks. Therefore, due to its stealthiness, sentence-level attack is a more challenging issue for adversarial defense in NLP. Yet, most of existing adversarial defenses (Jia et al., 2019; Dong et al., 2021; Wang et al., 2021a; Mozes et al., 2021; Wang et al., 2021b; Mosca et al., 2022; Le et al., 2022a) focus on defending word-level attacks, and they will also counter significant performance drop when facing other text attacks (e.g., sentence-level).

Second, there seems to be a contradiction between the generality and performance of a defense method. There are some general defenses (Le et al., 2022b; Liu et al., 2022) that can defend sentence-level, word-level and character-level attacks. However, their performance is not comparable to state-of-the-art defense methods designed for a specific attack (e.g., word-level attacks). Meanwhile, the success of state-of-the-art defense against one kind of attacks (e.g., word-level) can not transfer to another kind of attacks (e.g., sentence-level) (Zhang et al., 2020; Goyal et al., 2022). Thus, how to tackle such a trade-off between generality and performance remains a valuable question. We believe that designing more suitable detectors and correctors is a plausible direction since our saliency-based defense performs properly and shows strong potential to detect each kind of attacks. Specifically, there are two lines of future work to continue: (1) designing stronger detectors that perform well on three kinds of attacks, and (2) designing better correctors with general correction schemes.

