# OpenReview forum: "TextShield: Beyond Successfully Detecting Adversarial Sentences in text classification"
_ICLR.cc/2023/Conference — ICLR 2023 poster_

### Official Review · Reviewer_Yn8V · 2022-10-25

**Confidence:** 3
**Clarity, Quality, Novelty And Reproducibility:** There are a few grammatical errors an…
**Correctness:** 3
**Technical Novelty And Significance:** 3
**Empirical Novelty And Significance:** 3
**Recommendation:** 5

**Strength And Weaknesses:**

Strengths:
* The authors present a new method based on saliency maps.
* On a few datasets and attacks, the authors show that they can successfully detect and defend against adversarial samples.


Weaknesses:
* Some baseline comparisons are missing, for instance "Defense against Synonym Substitution-based Adversarial Attacks
via Dirichlet Neighborhood Ensemble" (Zhou et al., 2021) present DNE. From looking at the numbers in the DNE paper, it also seems like some cases performance of DNE is better.
* The authors mention that one of the weaknesses for adversarial training is that the robustness of the model is determined by the diversity of adversarial examples used in adversarial training and that you cannot defend against unknown attacks. Isn't the same thing true for TextShield as well because the detector needs to be trained with adversarial examples? Does TextShield generalize well to unknown attacks or to attacks that haven't been used during the training of the TextShield detector?
* The authors state that "there remains a significant gap between successfully detecting adversarial sentences and correctly classifying adversarial sentences for a specific application." This claim is confusing because defense methods are specifically designed for this.

**Summary Of The Paper:**

The authors propose a saliency based detector TextShield to identify adversarial samples in NLP. They use existing methods (vanilla gradient, guided backpropagation, layerwise relevance propagation, and integration gradient) to obtain saliency maps and then combine the outputs of all the methods to produce a final prediction of whether a sample is adversarial. The authors also introduce a corrector that corrects that modifies adversarial samples such that they are correctly classified.

**Summary Of The Review:**

I'm not entirely convinced about the empirical evaluation because there are a few missing baselines.

---

> ### Author Response · Authors · 2022-11-18
> **Response to reviewer Yn8V**
>
> We thank you for your previous time on our work, here’re our answers for your questions. Hopefully, they can alleviate your concerns.
>
> 1.**[Baseline selection]**: This is a good question that we did not explain in the paper: we didn't test DNE in our paper due to one principal reason: the source code of DNE is not released, so it is very difficult for us to reproduce their method, due to experimental settings between ours and theirs. Thus, it is also not fair to directly compare our results to the ones in their paper. Luckily, another concurrent work is very similar to DNE: ASCC because (a) DNE and ASCC both leverage the same method of adversarial optimization in a convex hull. (b) DNE and ASCC both encourage the sparsity of weights: DNE uses Dirichlet distribution, and ASCC uses entropy. Besides, ASCC releases its source code so that we can conduct experiments under the same setting. Therefore, our solution is to select ASCC as our baseline, and we believe ASCC is a more suitable representative method.
>
> 2.**[OOD issue in our method]**: Thanks for pointing out these critical issues: Empirically, our TextShield can generalize to the out-of-domain (OOD) attacks, so the success of TextShield is not very dependent on training with attacks used during evaluation. We conducted comprehensive experiments and concluded that TextShield could defend well against OOD word-level attacks, as shown in Appendix C. For OOD word-level attacks, TextShield still outperforms the baselines. Besides, we conduct detailed analyses to show why TextShield can generalize to unseen word-level attacks empirically. For OOD sentence-level attacks, all of the defenses against word-level attacks fail to defend well against sentence-level attacks. This result has been discussed in Appendix H in our original submission, a common issue in the adversarial NLP community ** how to tackle such a trade-off between generality on all attacks and performance against specific attacks remains a valuable question**, in the last paragraph in Appendix H.
>
> 3.**[Confusing statement]**: Sorry for some confusing points. The core limitation of previous detection methods is being incapable of giving correct predictions on adversarial sentences, unlike defense methods from other paradigms. Our paper aims to break this fatal limitation and solve this issue. We have made the statement less confusing in the updated version, as shown in the abstract.

---

### Official Review · Reviewer_iR5J · 2022-10-28

**Confidence:** 4
**Correctness:** 4
**Technical Novelty And Significance:** 3
**Empirical Novelty And Significance:** 2
**Recommendation:** 5

**Clarity, Quality, Novelty And Reproducibility:**

The paper is easy to understand. The authors have found a different way to defend against attacks.

**Strength And Weaknesses:**

Strengths:
1. This paper is well written, with a clear description and exact formula.
2. This paper aims to not only successfully detect adversarial sentences but also correctly classify adversarial sentences that previous attacking work rarely pays attention to. In order to enhance the robustness of victim models, previous work generally mixes the detected adversarial sentences and original training data to retrain the victim model. This work provides a new way to explore how to get correct predictions of those adversarial samples.
3. The experimental results on adversarial sentences produced by four widely-used text attacks in the NLP field achieve comparable performance to previous defense methods.

Weakness:
1. Some attackers get adversarial sentences by adding an imperceptible perturbation into benign sentences. Ideally, the adversarial sentence's target remains the same as the benign one. But the perturbation process can not promise the unchangeable targets of the adversarial sentences. The automatic evaluation can only take the targets of the benign sentences as the true targets of the adversarial sentences, but maybe they are not. So it is better to add a human evaluation on the part of samples to reflect the precision of Table 1.
2. This method is somewhat opportunistic. By adding a "shield", the victim model could be more likely to make correct predictions of adversarial samples, but there still have been some concerns. The first one is the speed of inference. Calculating saliency should go through a forward and a backward process of the victim model. So, I wonder about the speed of the detector and the corrector. Another is that the detector is trained on the adversarial sentences generated by the GA, IGA, PWWS, and TextFooler. The reported final experiment results are also on the adversarial sentences generated by the GA, IGA, PWWS, and TextFooler. So I wonder if the adversarial sample is "out of domain" (for example, train detector and test the "shield" using two different "styles" attacker), the performance of the detector. Such a situation is more realistic because an attacker may generate adversarial samples with the same "style", which can not cover all potential adversarial sentences.
3. The experiments have compared the saliency detector with previous detectors by adding the designed corrector to previous detector-based methods, with the results demonstrating the corrector's efficiency. I'd like to know how much of a role the saliency played in the corrector. Specifically, how will the model behave if I do not choose the replaced word according to the saliency but something else, like some entities? It is better to have some case studies to explicitly show the saliency scores and their importance.


**Summary Of The Paper:**

This paper proposed a detection-correction paradigm to defend against word-level attacks on the text classification task.
Their adversarial sentence detector leverages adaptive word importance (AWI) based on saliency computation. If a sentence is detected as an adversarial one, the corrector will act on the sentence by refining the word with the highest saliency score. Then, the victim model takes the corrected sentence as input to make the final prediction. Thus, their model is like a shield in front of the victim classifiers.

**Summary Of The Review:**

This paper proposed an interesting way of defense by adding a "shield" of the victim model. But there are some concerns like the inference speed, the generalization of the detector and the corrector. I will reconsider this paper if the authors can give some explanation and evidence of these problems.

---

> ### Author Response · Authors · 2022-11-18
> **Response to reviewer iR5J**
>
> Thanks for your inspiring comments on our paper, and we do appreciate your effort for reviewing our work. Here're our answers towards your questions:
>
> 1.**[human evaluation of attack quality]**: Thanks for pointing this out, this 'adversarial sentence quality' issue serves as a general problem which have been ommited by many SoTA defense methods. Specifically, considering the number of adversarial sentences and the labor brought by that, we select 200 samples for each attack, and manually evaluate their quality, and then report F1-score on these samples. The results are updated in Appendix E.
>
> 2.**[OOD issue contained in our method]**: Thanks for your valuable comments, our answers towards these questions are as follows:
> (1)	Considering efficiency, we provide a case study about computational cost in Appendix F, hope they can solve your concerns.
> (2)	 We conduct a detailed analysis towards the robustness of TextShield under OOD attacks, from where we can see the potential and advantages of TextShield. We add this part to Appendix C. Besides, our original version (Appendix H) also contains adversary detection performance comparisons between our saliency-based detector and baselines when facing OOD attacks, which also shows our detector's superiority.
>
> 3.**[baselines of our corrector design]**: This is an attractive question, thanks for pointing out. Since the baselines of our corrector in NLP remain blank, we define four intuitive word replacement strategies for corrector as follows: (1) replace word based on their part of speech (noun), which contains entity (2) replace word based on their part of speech (verb) (3) replace word based on their part of speech (noun+verb) (4)we replace word based based on their word frequency (in benchmark). The results are also updated in Appendix G.

---

### Official Review · Reviewer_xPPy · 2022-10-28

**Confidence:** 3
**Clarity, Quality, Novelty And Reproducibility:** please see the Strength And Weaknesse…
**Correctness:** 3
**Technical Novelty And Significance:** 3
**Empirical Novelty And Significance:** 2
**Recommendation:** 8

**Strength And Weaknesses:**

Strengths

- The authors present strong motivation in Section 4 about the connection between saliency and robustness
- The authors present a concise, clear description of adversarial text attacks and saliency computation.
- I like that TextShield has both a detector and a corrector for adversarial sentences which seems to be novel
- The proposed adaptive word importance (AWI) is interesting in that it is a simple, efficient method for identifying which words are the most important
-TextShield outperforms existing methods in adversarial detection metrics on popular datasets

Weaknesses
- Table 1 could be better presented if TS had (ours) in front of it to make it easy to distinguish between the proposed method and the rest. For the rest of the methods, it would be nice to have citations in front of them to make it easy to identify where they come from.
- Multiple runs of the experiments in Table 1 are missing as the variance based on the random seed would give us a good idea about whether the reported results are significant. It is not clear whether TS significantly outperforms others.
- No code is provided to reproduce the results and get a deeper understanding of the control flow.
- While this work doesn't seem entirely novel as the methods are based on popular computer vision algorithms related to saliency: the authors used saliency at the word level compared to the pixel level used in computer vision, which is very similar. However,  I like this work as it provides a simple yet efficient way of achieving strong performance compared to existing methods for adversarial detection and correction.

**Summary Of The Paper:**

The authors propose a method called TextShield to handle a problem setup within the field of adversarial attacks, where the goal is to first detect an adversarial sentence and then correctly classify it. The system detects adversarial sentences using a saliency-based methodology and subsequently corrects them to benign ones. TextShield surpasses existing approaches on multiple metrics in experiments on standard NLP datasets.

**Summary Of The Review:**

please see the Strength And Weaknesses section. While the method is not entirely novel, the method TextShield is an easy, efficient way to detect and correct adversarial attacks while achieving competitive results for NLP type of datasets.

---

> ### Author Response · Authors · 2022-11-18
> **Response to xPPy**
>
> We appreciate for your recognition for our paper, here're our replies that may alleviate your concern:
>
> 1.**[writings]**: Thanks for pointing our shortcomings in our writings. We've corrected corresponding places in current version for better clarification, as shown in Table 1.
>
> 2. **[table presentation and variance]**: We conducted experiments with three different random seeds, we specifically present the variance of our method and the best baseline SEM as follows:
> | Dataset | GA | IGA | PWWS | TextFooler | BA |
> |:---:|:---:|:---:|:---:|:---:|:---:|
> | TS(Ours) | **75.5(1.2)** | **76.5(1.0)** | **77.5(1.9)** | **75.0(1.3)** | **64.5(1.2)** |
> | SEM | 70.5(1.1) | 71.1(1.4) | 72.5(1.4) | 72.4(1.0) | 61.4(0.8) |.
>
> we can observe that TextShield outperforms SEM with statistical significance.
>
> 3. We are cleaning up the code and we can share it when it's ready.
>
> 4.We’ve updated many other interesting analyses and experiments in the current version, as shown in the general response. Please feel free if you have any questions about them.

---

### Official Review · Reviewer_V9s1 · 2022-10-29

**Confidence:** 3
**Correctness:** 3
**Technical Novelty And Significance:** 2
**Empirical Novelty And Significance:** 3
**Recommendation:** 5

**Clarity, Quality, Novelty And Reproducibility:**

Pls see Sec. Strength And Weaknesses


**Strength And Weaknesses:**

Strengths:

(1)	The illustration structure for intuitions and explanations for TextShield is stated clearly with the help of Figure 1-2.

(2)	The experimental part has conducted comprehensive ablation study on considering the impacts of the hyperparameter, multiple sub-detectors and the number of adversarial examples.



Weaknesses:

(1)	Overclaim in Section 2 (Related Work): "However, because of the extreme time cost in the training stage, certified robustness is difficult to be applied to complex models. For example, IBP performed catastrophically on large pre-trained language models (e.g., BERT)". Certified robustness of recent work can be applied to complex models like BERT [1].

(2)	In Subsection 5.3 (Corrector): "For a suspect w_{i}, the corrector substitutes w_{i} with the most frequent word w^{'} ∈ S(w_{i}), where S(w_{i}) is w_{i} 's synonyms [4]". Here the synonym candidate set is selected from NLTK toolkit. However, the adoption of synonym sets is different in research on adversarial attacks, and the defence method assumes that the defenders are aware of the strategy for adversaries to create synonyms. Because we cannot limit a malicious attacker's use of synonym tables, this scenario is not realistic.

(3)	In the experimental part, TL, PWWS, GA and IGA are selected as word-level attacks for generating adversarial examples. However, some SoTA attacks like TextFooler [2], and Bert-Attack [3] are not considered as baselines to test the effectiveness of the proposed defence method.

(4)	The computation runtime and efficiency of the proposed defence strategy are not discussed in the experimental part.


Typos:

(1)	"Given sentence X with label y_{j} predicted by text classifier F, the adaptive word importance (AWI) of word w_{i} in the sentence, R_{ij}, …" in Subsection 4.2 lacks articles. Such a problem also appears in several places in this paper.

(2)	"Takings VG as an example" in Subsection 5.2 (Detector).


Reference:

[1] Ye, Mao, Chengyue Gong, and Qiang Liu. "SAFER: A Structure-free Approach for Certified Robustness to Adversarial Word Substitutions." Proceedings of the 58th Annual Meeting of the Association for Computational Linguistics. 2020.

[2] Jin, Di, et al. "Is bert really robust? a strong baseline for natural language attack on text classification and entailment." Proceedings of the AAAI conference on artificial intelligence. Vol. 34. No. 05. 2020.

[3] Li, Linyang, et al. "BERT-ATTACK: Adversarial Attack Against BERT Using BERT." Proceedings of the 2020 Conference on Empirical Methods in Natural Language Processing (EMNLP). 2020.

[4] Loper, Edward, and Steven Bird. "Nltk: The natural language toolkit." arXiv preprint cs/0205028 (2002).


**Summary Of The Paper:**

This paper proposed a detection-based defence strategy called TextShield against word-level text attacks, incorporating a detector and a corrector. Four saliency computation methods, vanilla gradient (VG), guided backpropagation (GBP), layerwise relevance propagation (LRP), and integration gradient (IG), are selected as sub-detectors. Adaptive word importance (AWI) is designed using the vanilla gradient method. Then a saliency-based corrector aims to correct adversarial sentences to benign ones based on AWI. Comprehensive experiments demonstrate that TextShield is superior to previous defence methods on both generalization and robustness.

**Summary Of The Review:**

Pls see Sec. Strength And Weaknesses

---

> ### Author Response · Authors · 2022-11-18
> **Response to Reviewer V9s1**
>
> 1. **[overclaim]**: Thanks for pointing out this. Sorry for such a misleading statement, we want to express that certified robustness is more difficult to apply on pre-trained model compared to other paradigms. When saying "performed catastrophically", we refer it to IBP instead of all certified robustness methods. We’ve made modification in the updated version, as shown in Sec 2.
>
> 2. **[synonym strategy]**: That's a good question. In our implementation, we fix the simple API from nltk.corpus.wordnet as our synonym creation strategy. Although we use this toolkit, this doesn’t mean we need to know the synonym creation strategy of text attacks. In our paper, GA, IGA, PWWS and TextFooler define synonyms by l2 distance between words’ embedding, which is different from our implementation. Thus, there’s no overlap between our synonym strategy and attacks’. Besides, we add a new word-level attack called BERT-attack, which replaces words by perplexity instead of synonym, our TextShield still outperforms the baselines. Overall, our synonym strategy is free of assumptions towards the strategies in attacks.
>
> 3. **[results from more attacks]**: The results of TextFooler have been included in the original submission in Table 2, where TL refers to TextFooler. We'll highlight this in the next version. Besides, we add the results of BERT-Attack as follows, which has been updated in Appendix D.
> | Dataset | NT | IBP | FGWS | WDR | ASCC | SEM | TS |
> |:---:|:---:|:---:|:---:|:---:|:---:|---|---|
> | IMDb | 10.8 | - | 50.2 | 59.0 | 61.4 | 61.4 | **64.5** |
> | AGNews | 9.6 | - | 46.7 | 52.4 | 60.2 | 59.6 | **63.0** |
> | Yahoo | 6.0 | - | 40.2 | 50.1 | 58.9 | 60.3 | **64.2** |
>
> 4. **[computational cost]**: Concerning the computational cost of our method, we present a case study here.
> We choose one RTX3090 GPU, and select BERT-base-uncased as the victim model and set the batch size as 8. Each inference of BERT takes around 20-30ms, TextShield (only detector part) takes around 60ms, and the cost is acceptable. The corrector + BERT inference takes around 20-30ms, nearly equal to BERT inference time since the information corrector needed has already been computed in the VG detector. Overall, it takes around 80-90ms per inference time under the configuration. Considering the training cost of detectors, since detectors only consist of LSTM and MLP, training them until convergence takes around 60min (On RTX3090). The contents are updated in Appendix F.

---

### Decision · Program_Chairs · 2023-01-20

**Decision:**

Accept: poster

**Justification For Why Not Higher Score:**

Besides the writing issue, the paper only considers attacking text classification models. Therefore, the scope of the study is narrow.

**Justification For Why Not Lower Score:**

There is no significant issue with the paper. The novelty of the proposed approach is moderate.

**Metareview: Summary, Strengths And Weaknesses:**


The paper presents an approach to defending against word-level adversarial attacks on text classification tasks. The method first detects adversarial sentences based on a saliency-based detector. Then a corrector is proposed to rewrite and fix the adversarial samples.

Overall, all reviewers agree that the paper is on the borderline and suggest accepting the paper if there is space. Specifically, the ideas presented in the paper are interesting and the rebuttal clarifies the technical concerns. The writing is admissible, but the authors are strongly encouraged to improve it

Strength:

+ Overall, the reviewers agree the proposed approach is novel and meets the bar of publishing at ICLR.

+ The experiments are comprehensive and support the claims.

+ The rebuttal resolves all the technical concerns.

Weakness:

- Parts of the paper remain unclear or misleading after the rebuttal. For example, the design of the corrector has to be clarified. The motivation and intuition behind the design choices should be elaborated. Overall, the writing is admissible but the authors are strongly encouraged to improve it. Please refer to the writing suggestions in the reviews for details.

- The paper only considers text classification tasks and it’s unclear if the proposed approach can be extended to other NLP tasks with complex structures such as question answering. I understand the majority of the adversarial attack and defense approaches for NLP, especially those published in ML venues, only consider text classifications, and the authors follow the literature to conduct experiments on the selected datasets. However, NLP is beyond text classification. The authors should clarify this in the writing and I would suggest changing the title to “TextShield: Beyond Successfully Detecting Adversarial Sentences in Text Classification”

Missing references:
Although the setup is different, the idea of identifying word-level adversarial attacks and patching the example for defense has been discussed in the following paper.
Learning to Discriminate Perturbations for Blocking Adversarial Attacks in Text Classification, EMNLP 19





**Note From Pc:**

if the above contains the word "oral" or "spotlight" please see: "oral" presentation means -> notable-top-5% and "spotlight" means -> notable-top-25%. As stated in our emails, we are disassociating presentation type from AC recommendations

**Summary Of Ac-Reviewer Meeting:**

All reviewers respond to the discussion requests. However, due to reviewers spreading out in more than 3 time zones. There is no way to schedule a meeting time that is reasonable for everyone. In the end, reviewers V9s1, xPPy, and Yn8V attend the virtual meeting and iR5J has to miss the meeting as the meeting time is 4am in iR5J’s local time. However, iR5J discussed the paper with the AC offline and agreed on the conclusion in the discussion.  Overall, all reviewers agree with each other on the pros and cons of the paper. Reviewer xPPy is more positive about the paper and the other three reviewers are more conservative about accepting the paper due to the writing. However, they all agree that the issues are relatively minor and can be fixed in the camera-ready. There is a long discussion about the role of the corrector proposed in the paper as the description is not clear to the readers even after rebuttal. The AC also checked the paper and agrees with the reviewers on the writing issue.

The meeting concludes leaning toward accepting the paper.